# Inverted Papilloma of the Maxillary Sinus: A Recurrence Analysis According to Surgical Approaches

**DOI:** 10.3390/jcm11113020

**Published:** 2022-05-27

**Authors:** Jin Youp Kim, Su Hwan Kim, Pilkeun Jang, Yuju Seo, Sung-Woo Cho, Jeong-Whun Kim, Doo Hee Han, Hyun Jik Kim, Dong-Young Kim, Chae-Seo Rhee, Tae-Bin Won

**Affiliations:** 1Department of Otorhinolaryngology-Head and Neck Surgery, Ilsan Hospital, Dongguk University, Goyang 10326, Korea; kjyoup0622@gmail.com; 2Interdisciplinary Program of Medical Informatics, Seoul National University College of Medicine, Seoul 03080, Korea; 3Department of Biomedical Engineering, Seoul National University College of Medicine, Seoul 03080, Korea; suhwan@ualberta.ca; 4Department of Otorhinolaryngology-Head and Neck Surgery, Seoul National University Bundang Hospital, Seongnam 13620, Korea; jpg0923@gmail.com (P.J.); iamsungu@gmail.com (S.-W.C.); kimemails7@gmail.com (J.-W.K.); 5Department of Otorhinolaryngology-Head and Neck Surgery, School of Medicine, Chungnam National University, Daejeon 35015, Korea; yuz1121@naver.com; 6Department of Otorhinolaryngology-Head and Neck Surgery, Seoul National University Hospital, Seoul 03080, Korea; handh@snu.ac.kr (D.H.H.); hyunjerry@snu.ac.kr (H.J.K.); dongkim@snu.ac.kr (D.-Y.K.); csrhee@snu.ac.kr (C.-S.R.)

**Keywords:** maxillary sinus, papilloma, inverted, paranasal sinuses

## Abstract

**(1) Background**: Various surgical approaches have been introduced to resect inverted papillomas (IP) stemming from the maxillary sinus (MS). This study aimed to compare the recurrence rates of IPs originating from the MS according to various surgical modalities. **(2) Methods**: A total of 155 surgical cases of sinonasal IPs originating from the MS were categorized into three groups according to the surgical approach adopted: endoscopic resection via middle or inferior meatus antrostomy (ESS), ESS with Caldwell–Luc approach or canine fossa trephination (ESS with CL), and expanded endoscopic approaches (ExEA) including endoscopic medial maxillectomy or a prelacrimal recess approach. A Kaplan–Meier curve was generated to examine the recurrence rates. **(3) Results**: The overall recurrence rate was 5.8% (9/155) with a mean follow-up period of 24.2 months. The recurrence rates for the ESS, ESS with CL, and ExEA groups were 10.0% (7/70), 3.5% (2/57), and 0% (0/28), respectively. The ExEA group had a significantly lower recurrence rate than the ESS group (*p* = 0.024) and there was a tendency for lower recurrence compared to the ESS within the CL group (*p* = 0.145). The ExEA required a shorter postoperative hospitalization period than in ESS with CL (*p* < 0.001). **(4) Conclusions**: ExEAs to the maxillary sinus such as the PLR and EMM approaches are excellent surgical options for IPs originating from the MS.

## 1. Introduction

Inverted papilloma (IP) of the nasal cavity and paranasal sinuses is a benign tumor that accounts for 0.5–4% of all sinonasal tumors [1]. IP has a high propensity for recurrence after surgery (5–30%) and an association with squamous cell carcinoma (5–21%) [2]. Therefore, a complete and meticulous resection is required to ensure a successful outcome. The maxillary sinus (MS) is the most common origin site for IPs, and various surgical approaches have been introduced to resect IPs stemming from the MS [3,4,5]. Common surgical modalities include endoscopic resection via middle meatus antrostomy (MMA), sometimes augmented by inferior meatus antrostomy (IMA) and/or mega-antrostomy. However, areas such as the alveolar recess and anterior and lateral walls of the MS are difficult to access with these approaches. The Caldwell–Luc approach via a standard sublabial incision can be combined with an endoscopic resection to overcome these limitations, providing unimpeded exposure of the MS [6,7]. Although the combined approach is effective, complications such as facial swelling and facial and dental numbness are frequent, presenting in up to 75% of patients [8,9].

Recently, the endoscopic approaches to the MS have evolved to include expanded procedures that can enhance visualization and surgical access to the MS. These so-called expanded endonasal approaches (ExEA) include the endoscopic medial maxillectomy (EMM) and the prelacrimal recess approach (PLR). Endoscopic medial maxillectomy (EMM) [3] provides wide access to the MS and facilitates complete resection; however, atrophic rhinitis and obstruction of the lacrimal pathway are potential complications of this approach as the inferior turbinate (IT) and nasolacrimal duct (NLD) are resected [3,10]. The PLR [11] provides a comparable wide exposure of the MS, including its anterior and lateral walls, while preserving the IT and NLD. Previous studies have compared the surgical outcomes using the PLR and conventional endoscopic approaches [7,12]; however, these studies had limitations in that they did not consider the wide variety of endoscopic approaches. This study aimed to compare the recurrence rates for IPs originating from the MS according to various surgical approaches.

## 2. Materials and Methods

### 2.1. Study Design and Participants

We conducted a retrospective analysis of patients who underwent surgery for sinonasal IPs originating from the MS between January 2011 and June 2020 at two tertiary medical centers. Sinonasal IP cases were categorized into four groups based on the surgical approach performed: endoscopic resection via MMA and/or IMA (ESS), ESS with the Caldwell–Luc approach, or canine fossa trephination (ESS with CL), and ExPA such as EMM or PLR. From the medical records, patient demographics, clinical characteristics including previous surgical history, and tumor attachment sites were extracted for the analysis. The Krouse staging system was used to evaluate the IP status [13]. The excised tissue was evaluated through postoperative pathologic examination and confirmed as an IP; dysplasia in the excised tissue was also assessed. Patients with concomitant squamous cell carcinoma were excluded. This study was performed in accordance with the Declaration of Helsinki. The present study was approved by the Institutional Review Boards of the Seoul National University Hospital and Seoul National University Bundang Hospital (IRB No. H-2102-033-1194 and B-2103/675-401, respectively). The need for informed consent was waived by the Institutional Review Board of Seoul National University Hospital and Seoul National University Bundang Hospital due to the retrospective nature of the study.

### 2.2. Expanded Endoscopic Approaches

Endoscopic medial maxillectomy can be performed in the usual fashion [3] by resecting medial wall of the MS, including the inferior turbinate and the nasolacrimal duct. A modified EMM can be performed, preserving the inferior turbinate. This is accomplished by maintaining the posterior attachment during resection of the IP in the MS and suturing the anterior end to the lateral wall at the conclusion of the surgery (Figure 1). For the prelacrimal recess approach, a curved incision is performed immediately posterior to the mucocutaneous junction, from the lateral wall of the nasal cavity to the nasal floor (Figure 2A). The mucoperiosteum is carefully elevated, identifying the following structures: bony pyriform aperture, lateral wall, anterior end of the IT, inferior meatus, and nasal floor (Figure 2B). A prelacrimal bone osteotomy is performed in the lateral wall with a straight 3 mm osteotome between the pyriform aperture and the IT attachment (Figure 2C). The NLD is easily identified while medializing the bony lateral wall of the nasal cavity (Figure 2D). The NLD can be further exposed by partial removal of the bony lateral wall. The lateral wall flap, consisting of the NLD and IT, is tented medially, exposing the MS medial wall (Figure 2E). A temporary suture of the lateral wall flap facilitates subsequent instrumentation passage through this corridor. The MS bony medial wall is removed, and the mucosa incised, allowing endoscopic IP examination and manipulation in the MS (Figure 2F). The corridor can be enlarged posteriorly to the MS posterior wall, superiorly to the lacrimal sac, and inferiorly to the nasal floor. The lateral nasal wall can be removed up to the pyriform aperture to reach the anterior and lateral walls of the MS. The tumor in the MS is debulked, isolating the attachment site (Figure 2G,H), which is resected with 1 cm margins using a multilayer centripetal technique (Figure 2I). The hyperostotic attachment site is usually drilled with preservation of the posterior periosteum (Figure 2J). Wide MMA is performed for endoscopic observation during follow-up (Figure 2K). After completing the resection, the lateral wall flap is repositioned using VICRYL sutures (Ethicon Inc., Bridgewater, NJ, USA) (Figure 2L).

### 2.3. Statistical Analysis

Continuous variables, such as demographic information, follow-up duration, and postoperative hospitalization days, are presented as the mean ± standard deviation; the Kruskal–Wallis test was used to compare the demographics between the four groups. Fisher’s exact test was used to evaluate the categorical variables. A Kaplan–Meier curve was generated to examine the recurrence rate, and a log-rank test was performed to compare the overall difference in recurrence rates between the four groups. The log-rank test was conducted at the point where all patients in one group were censored. For this reason, we performed a fixed time-point survival test [14] developed by Klein et al. to assess the difference in recurrence rates at a certain time point (t = 74.2 months, when the last ExEA patient was censored) as post hoc analysis. For multiple comparisons, the Benjamini and Hochberg procedure [15] was performed. All statistical analyses were conducted using R for Windows version 4.0.3 (R Foundation for Statistical Computing, Vienna, Austria). A *p*-value < 0.05 was set as the threshold for statistical significance.

## 3. Results

In total, 155 cases (148 patients, including 7 patients who had surgery twice due to IP recurrence) of IP originating from the MS were analyzed in this study. The mean patient age was 55.9 years, and the mean follow-up duration was 24.2 months. Among the 155 cases, 117 were primary and 38 were revision surgeries. MS IP cases were categorized into three groups based on the surgical approaches adopted: ESS (n = 70), ESS with CL (n = 57), and ExEA (n = 28). Demographics and clinical characteristics of the four groups of patients are presented in Table 1. The rate of revision cases was significantly higher in the ESS with CL group than the ESS or ExEA group (*p* = 0.004 and *p* = 0.005, respectively), as wide MS exposure is commonly required in revision cases. Although the ESS group had a shorter follow-up duration than the ExEA or ESS with CL groups (*p* = 0.008 and *p* < 0.001, respectively), the follow-up duration in the ExEA and ESS with CL group was comparable (*p* = 0.275). The hospitalization period in the ESS with CL group was significantly longer than in the ESS or ExEA groups (both *p* < 0.001).

The overall recurrence rate was 5.8% (9/155), and the rates for the ESS, ESS with CL, and ExEA groups were 10.0% (7/70), 3.5% (2/57), and 0% (0/28), respectively. The Kaplan–Meier plot showed a significant difference between the three groups (*p* = 0.004; Figure 3). In the post-hoc analysis, the ExEA group had a significantly lower recurrence rate than the ESS group (*p* = 0.024). Although statistical significance was not observed, the recurrence rate in the ExEA group was lower compared to the ESS with CL group (*p* = 0.145). These results suggest superiority of the ExEA approach for IPs originating in the MS, compared to ESS, and at least a comparable efficacy to ESS with CL.

The recurrence rates in primary and revision cases were 6.0% (7/117) and 5.3% (2/38), respectively. The Kaplan–Meier plot revealed no significant difference in recurrence rates between primary and revision cases (Figure 4). In a subgroup analysis excluding revision cases (n = 117), the recurrence rate in the ExEA group was significantly lower than that in the ESS group (*p* = 0.010; Figure 5), which was consistent with the analysis of all patients.

The origin sites within the MS are shown in Table 2. Overall, the superior wall of the MS was the most common origin site (25.2%), followed by the medial and anterior walls (both 20.0%). The superior wall was the most common origin site for the ESS and ExEA groups, with 28.6% and 25.0% of cases, respectively. In the ESS with CL group, the anterior wall was the most common origin site (28.1%). The highest recurrence rate based on the origin site within the MS was for the lateral wall (15.0%), followed by the medial and posterior walls (9.7% and 7.1%, respectively; Table 3). All recurrences originating from the lateral wall of the MS occurred in the ESS group.

## 4. Discussion

Resection of IPs originating from the MS can be performed in various ways. A traditional medial maxillectomy via a sublabial approach accomplishes an en bloc resection of the lateral wall of the nasal cavity, including the MS medial wall. Although effective, this technique is also associated with complications [10]. In a previous study of 35 patients who underwent a medial maxillectomy, all participants reported prolonged nasal crusting, one of the symptoms of atrophic rhinitis [10]. In addition, the reported incidence of epiphora after this surgery is approximately 30% [3].

With the advances in endoscopic sinus surgery, endoscopic resection has largely replaced the more invasive external surgical procedures. However, recurrence is not infrequent, as shown in a multicenter study including 578 patients, where the recurrence rate of IP originating from the MS was 12.9–21.6% [16]. This finding has urged surgeons to combine external approaches, such as the Caldwell–Luc approach or canine trephination, for better access to the MS, especially for lesions located in the anterior and/or lateral walls. However, combined approaches are inherently associated with prolonged hospitalization and surgical complications, such as facial swelling and numbness.

EMM [3] can replace the external approach while achieving similar results. However, as the IT and NLD are sacrificed, atrophic rhinitis and obstruction of the lacrimal pathway may occur, and modifications that can preserve the IT have been introduced [3,10,11]. There is growing evidence of the efficacy of PLR [5,12,17,18,19], which allows comparable access to the MS as the EMM while preserving the IT and NLD. A similar recurrence rate between the PLR and Caldwell–Luc approaches has been reported [12]. However, the possible superiority of one technique compared to other endoscopic approaches has not been thoroughly evaluated [7].

In the present study, we compared the recurrence rate of IPs in the MS after surgical resection via the ExEA to ESS and ESS with CL. The overall recurrence rate of 5.8% in our study is lower than in a previous multicenter study (12.9–21.6%) [16]. The recurrence rate in the ExEA group was significantly lower than that in the ESS group; no recurrence was found in the ExEA group, suggesting the superiority of ExEA to conventional ESS, which includes excision via MMA and/or IMA. It also showed a tendency of a lower recurrence compared to the combined approach (ESS with CL or trephination) with a comparable follow-up period.

Although access to the MS can be improved with the addition of the Caldwell–Luc approach or canine trephination, a thorough visualization of the attachment site or sites within the MS is challenging, even with angled endoscopes leaving possible blind spots. On the contrary, the anterior and lateral walls of the MS can be widely visualized and exposed through the ExEAs. Instrumentation can be facilitated by adding a saucerization procedure, which involves removing the lateral wall of the nasal cavity up to the pyriform aperture. Preserving this structure can avoid long-term changes in the shape of the alar base. The alveolar recess can also be accessed by lowering the medial wall of the MS up to the nasal floor. For instrumentation of lesions located in the far lateral aspect of the MS, a contralateral transseptal approach can facilitate resection. We believe that these surgical techniques have collectively contributed to the absence of recurrence in the ExEA group, which also included patients whose IP involved the lateral and or anterior walls. Furthermore, the wide surgical view through ExEA can facilitate demucosation and bone removal if needed, the key procedure for preventing recurrence, compared to ESS and ESS with CL.

The ExEA group included the conventional EMM where the nasolacrimal duct and inferior turbinate are resected, the modified EMM where the IT was preserved, and the PLR. As previously mentioned, EMM provides wide access to the MS which is comparable or even better than the PLR. While there was no patient with nasolacrimal duct obstruction in the PLR group, one patient developed epiphora after EMM and was subsequently treated with an endoscopic dacryocystorhinostomy. Presently, there are no patients who complain of symptoms related to empty-nose syndrome, such as excessive crusting dryness and pain. Future studies with larger patient numbers and longer follow-up will enable us to determine a clinically meaningful difference in complications.

It is well documented that revision surgery for IPs carries a higher rate of recurrence than primary surgery, as the site of origin may not be accurately localized due to postoperative tissue changes [20]. Furthermore, the recurred cases may have a different tumor biology that naturally increases their risk of recurrence [20]. However, we did not observe a significant difference in recurrence rate between primary and revision cases. This finding may be partially explained by the inclusion criteria of this study, which were limited to IPs originating from the MS and excluded Krouse stage 4 cases. Considering the difference in the ratio of primary and revision cases between groups, we performed a subgroup analysis for the primary cases and obtained consistent results.

This study has some limitations. The number of patients in the ExEA group is relatively small compared to cases of ESS or ESS with CL. In addition, although the attachment site of the IP within the maxillary sinus was investigated revealing the highest recurrence for the lateral wall, the retrospective nature of the study limits a meaningful interpretation according to the surgical approach. Lastly, despite the mean follow-up duration of 24.2 months, the recurrence rate with longer follow-up should be evaluated in future studies. In spite of these limitations, the strength of this study is that it compared the recurrence rates of IPs originating from the MS resected through a variety of surgical approaches.

In conclusion, for IPs originating from the MS, the recurrence rate after endoscopic resection with the ExEA was lower than that after ESS and similar to that after ESS with CL, although the latter required longer postoperative hospitalization. The ExEA is an excellent option for complete resection of IPs originating from the MS.

## Figures and Tables

**Figure 1 jcm-11-03020-f001:**
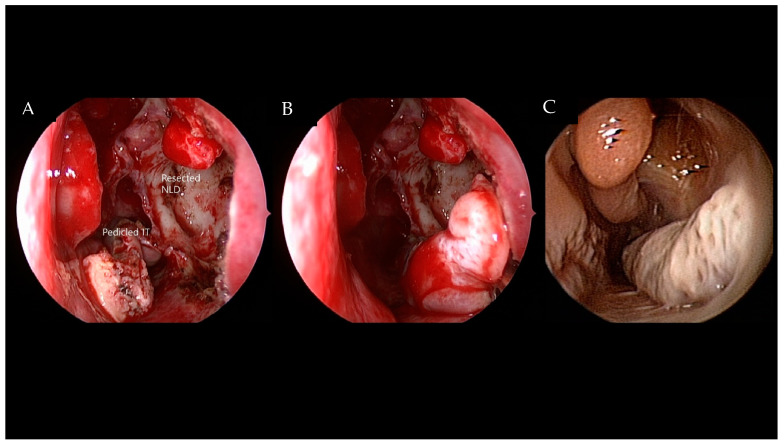
**Modified endoscopic medial maxillectomy approach with preservation of the inferior turbinate.** (**A**) Lacrimal duct has been resected. The inferior turbinate is pedicled posteriorly providing a wide access to the left maxillary sinus. (**B**) After complete removal of the inverted papilloma, the inferior turbinate is repositioned. (**C**) Postoperative 1-year endoscopic view of the left nasal cavity with well-preserved inferior turbinate. IT—inferior turbinate; NLD—nasolacrimal duct.

**Figure 2 jcm-11-03020-f002:**
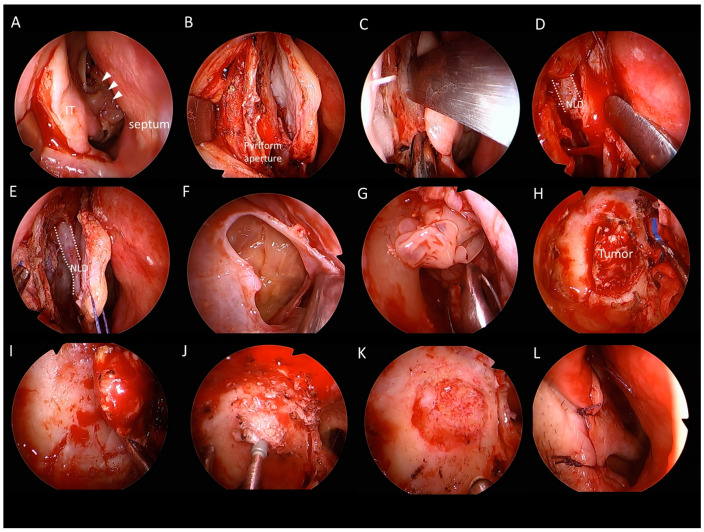
**Prelacrimal recess approach in a patient with an inverted papilloma in the right maxillary sinus.** (**A**) Curved incision from the lateral wall of the nasal cavity to the nasal floor. The arrows indicate inverted papilloma. (**B**) Bony pyriform aperture, lateral wall, anterior end of the IT, inferior meatus, and nasal floor are identified. (**C**) Prelacrimal bone osteotomy. (**D**) The NLD is easily identified while medializing the bony lateral wall of the nasal cavity. (**E**) The lateral wall flap consisting of the NLD and IT is tented medially, exposing the medial wall of the MS. (**F**) The bony medial wall of the MS is removed, and the mucosa incised. (**G**,**H**) The tumor in the MS is debulked, isolating the attachment site. (**I**) Multilayer centripetal resection with 1 cm margins. (**J**) Drilling the hyperostotic attachment site. (**K**) Wide MMA. (**L**) Repositioning the lateral wall flap. IT—inferior turbinate; MMA—middle meatal antrostomy; MS, maxillary sinus; NLD, nasolacrimal duct.

**Figure 3 jcm-11-03020-f003:**
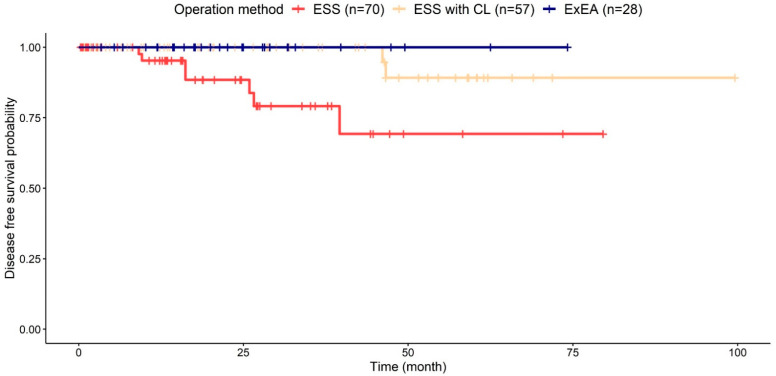
**Kaplan–Meier plot according to surgical approach.** There were nine recurrences in total: ESS, 10.0% (7/70); ESS with CL, 3.5% (2/57); and EAA, 0% (0/18). There was a significant difference in recurrence rate between the three groups (*p* = 0.004, log-rank test). The ExEA group had a lower recurrence rate than the ESS group at 74.2 months after the surgery (*p* = 0.024, fixed time-point survival test). ExEA—expanded endoscopic approach; ESS—endoscopic resection via middle meatal antrostomy and/or inferior meatal antrostomy; ESS with CL—ESS with the Caldwell–Luc approach or canine fossa trephination.

**Figure 4 jcm-11-03020-f004:**
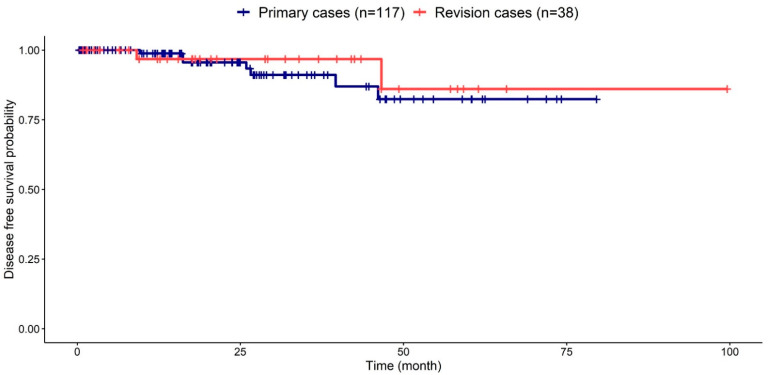
**Kaplan–Meier plot for primary and revision cases.** There were seven and two recurrences in primary and revision cases, respectively. There was no significant difference in recurrence rates between the two groups (*p* = 0.600, log-rank test).

**Figure 5 jcm-11-03020-f005:**
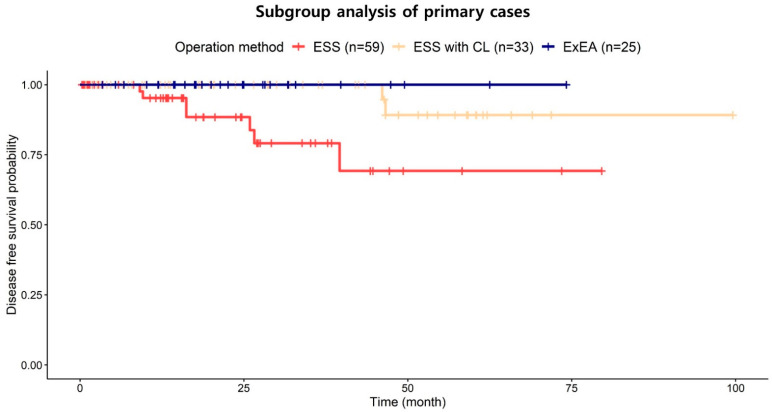
**Kaplan–Meier plot for the subgroup analysis of primary cases.** There were seven recurrences among the primary cases: ESS, 10.2% (6/59); ESS with CL, 3.0% (1/33); and ExEA, 0% (0/25). There were significant differences in recurrence rates between the three groups (*p* = 0.010, log-rank test). The ExEA group had a lower recurrence rate than the ESS group at 74.2 months after surgery (*p* = 0.035, fixed time-point survival test). ExEA—expanded endoscopic approach; ESS—endoscopic resection via middle meatal antrostomy and/or inferior meatal antrostomy; ESS with CL—ESS with the Caldwell–Luc approach or canine fossa trephination.

**Table 1 jcm-11-03020-t001:** Demographic and clinical characteristics of the patients.

	ESS	ESS with CL	ExEA	*p*-Value
	n = 70	n = 57	n = 28	
Age, year	55.6 ± 12.6	55.5 ± 12.0	57.4 ± 11.5	0.540 ^†^
Sex, male/female	48/22	35/22	18/10	0.716 ^‡^
Revision cases	11/70 (15.7%)	24/57 (42.1%)	3/28 (10.7%)	<0.001^‡^
Follow-up, months	17.4 ± 17.7	31.9 ± 23.1	25.6 ± 16.7	<0.001 ^†^
Krouse staging				0.403 ^‡^
T1	0	0	0	
T2	7	4	2	
T3	63	53	26	
T4	0	0	0	
Dysplasia	4/70 (5.7%)	1/57 (1.8%)	1/28 (3.6%)	0.655 ^‡^
Postoperative hospitalization, days	1.1 ± 0.6	1.7 ± 0.8	1.0 ± 0.3	<0.001 ^†^

The values are demonstrated as mean ± standard deviation. *p*-values were calculated using the ^†^ Kruskal–Wallis test and ^‡^ Fisher’s exact test. Abbreviations: ExEA—expanded endoscopic approach; ESS—endoscopic resection via middle meatus antrostomy and/or inferior meatus antrostomy; ESS with CL—ESS with the Caldwell–Luc approach or canine fossa trephination.

**Table 2 jcm-11-03020-t002:** Origin site in the maxillary sinus.

	Total Cases	ESS	ESS with CL	ExEA
	N = 155	N = 70	N = 57	N = 28
Anterior wall	31 (20.0%)	9 (12.9%)	16 (28.1%)	6 (21.4%)
Posterior wall	14 (9.0%)	6 (8.6%)	4 (7.0%)	4 (14.3%)
Lateral wall	20 (12.9%)	13 (18.6%)	3 (5.3%)	4 (14.3%)
Medial wall	31 (20.0%)	15 (21.4%)	11 (19.3%)	5 (17.9%)
Superior wall	39 (25.2%)	20 (28.6%)	12 (21.1%)	7 (25.0%)
Inferior wall	15 (9.7%)	7 (10.0%)	7 (12.3%)	1 (3.6%)
Diffuse origin *	5 (3.2%)	0 (0.0%)	4 (7.0%)	1 (3.6%)

Abbreviations: ExEA—expanded endoscopic approach; ESS—endoscopic resection via middle meatus antrostomy and/or inferior meatus antrostomy; ESS with CL—ESS with the Caldwell–Luc approach or canine fossa trephination. * Inverted papilloma originating from multiple sites involving two or more walls of the maxillary sinus.

**Table 3 jcm-11-03020-t003:** Recurrence rates according to the origin site in the maxillary sinus.

	Total Cases	ESS	ESS with CL	ExEA
	N = 155	N = 70	N = 57	N = 28
Anterior wall	3.2% (1/31)	11.1% (1/9)	0.0% (0/16)	0.0% (0/6)
Posterior wall	7.1% (1/14)	16.7% (1/6)	0.0% (0/4)	0.0% (0/4)
Lateral wall	15.0% (3/20)	23.1% (3/13)	0.0% (0/3)	0.0% (0/4)
Medial wall	9.7% (3/31)	6.7% (1/15)	18.2% (2/11)	0.0% (0/5)
Superior wall	2.6% (1/38)	5.0% (1/20)	0.0% (0/12)	0.0% (0/7)
Inferior wall	0% (0/15)	0% (0/7)	0.0% (0/7)	0.0% (0/1)
Diffuse origin *	0% (0/5)	NA	0.0% (0/4)	0.0% (0/1)
Overall	5.8% (9/155)	10.0% (7/70)	3.5% (2/57)	0.0% (0/28)

Abbreviations: ExEA—expanded endoscopic approach; ESS—endoscopic resection via middle meatus antrostomy and/or inferior meatus antrostomy; ESS with CL—ESS with the Caldwell–Luc approach or canine fossa trephination; NA, not available. * Inverted papilloma originating from multiple sites involving two or more walls of the maxillary sinus.

## Data Availability

Data are available upon reasonable request.

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
