# Peer review of "Inverted Papilloma of the Maxillary Sinus: A Recurrence Analysis According to Surgical Approaches"

_jcm, 2022, doi:10.3390/jcm11113020_

Round 1

Reviewer 1 Report

This is a well-written manuscript regarding the recurrence rates of various surgical approaches for removing inverted papillomas originating from the maxillary sinus.

Only several  comments are suggested.

  1. The abbreviation of expanded endoscopic approaches (ExEA or EEA) should be consistent in the whole manuscript. (Figure 5 legend, Table 2, Table 3)
  2. No long-term follow-up data may be listed as one of the limitations in the Discussion part.
  3. In the Results section, paragraph 1, line 5~6, it should be "three" groups instead of " four" groups.
  4. Demucosation may be the key procedure for preventing recurrence. ExEA provides better surgical view for demucosation comparing to ESS and ESS with CL. 

Author Response

Point 1. The abbreviation of expanded endoscopic approaches (ExEA or EEA) should be consistent in the whole manuscript. (Figure 5 legend, Table 2, Table 3)

Response 1: Thank you for the comment. We have changed all “EEA” to “ExEA” in the entire manuscript.

Point 2: No long-term follow-up data may be listed as one of the limitations in the Discussion part.

Response 2: We appreciate your insight. With respect to your comment, we have added a sentence in the limitation section as follows:

“…… In addition, although the attachment site of the IP within the maxillary sinus was investigated revealing the highest recurrence for the lateral wall, the retrospective nature of the study limits a meaningful interpretation according to the surgical approach. Lastly, despite the mean follow-up duration of 24.2 months, the recurrence rate with longer follow-up should be evaluated in future studies.”

Point 3: In the Results section, paragraph 1, line 5~6, it should be "three" groups instead of " four" groups.

Response 3: We appreciate the comment. We have changed the sentence as follows:

“MS IP cases were categorized into three groups based on the surgical approaches adopted: ESS (n=70), ESS with CL (n=57), and ExEA (n=28).”

Point 4: Demucosation may be the key procedure for preventing recurrence. ExEA provides better surgical view for demucosation comparing to ESS and ESS with CL.

Response 4: We appreciate your remark. With respect to your comment, we have added a sentence in the discussion section as follows:

“…… We believe that these surgical techniques have collectively contributed to the absence of recurrence in the ExEA group which also included patients whose IP involved the lateral and or anterior walls. Furthermore, the wide surgical view through ExEA can facilitate demucosation and bone removal if needed, the key procedure for preventing recurrence, compared to ESS and ESS with CL.”

Reviewer 2 Report

Difference between the sample number of the surgical techniques

Author Response

Point 1. Difference between the sample number of the surgical techniques

Response 1: We appreciate your remark. As expanded endoscopic approaches (ExEA) is currently developed procedure, the sample size of ExEA is relatively small. Therefore, we mentioned it in the limitation section (“The number of patients in the ExEA group is relatively small compared to cases of ESS or ESS with CL.”). The authors think this study validates the efficacy of ExEA, which facilitate usage of ExEA for inverted papilloma in maxillary sinus, and further study can be performed with a sufficient number of ExEA cases.

Reviewer 3 Report

The paper shows a significant variation in a minimally invasive surgical approach to the maxillary sinus. It is interesting.

Author Response

Point 1. The paper shows a significant variation in a minimally invasive surgical approach to the maxillary sinus. It is interesting.

Response 1: We appreciate the comment.

Reviewer 4 Report

Different surgical approaches for the management of maxillary inverted papilloma were compared, and the superiority of the endoscopic prelacrimal approach in terms of the recurrence rate was shown. Although the number of each group was not very big, the shown result was reasonable.

line 103, "en bloc".

The shown removal method is not en bloc, but multilayer resection or multilayer centripetal resection.

Author Response

Response to Reviewer 4 Comments

Point 1. Different surgical approaches for the management of maxillary inverted papilloma were compared, and the superiority of the endoscopic prelacrimal approach in terms of the recurrence rate was shown. Although the number of each group was not very big, the shown result was reasonable.

Response 1: We appreciate the comment.

Point 2. line 103, "en bloc". The shown removal method is not en bloc, but multilayer resection or multilayer centripetal resection.

Response 2: Thank you for the comment. We changed “en bloc resection” to multilayer centripetal resection as follows:

“The tumor in the MS is debulked, isolating the attachment site (Fig.2G, 2H) which is resected with 1-cm margins using a multilayer centripetal technique (Fig.2I).”

“Figure 2. Prelacrimal recess approach in a patient with an inverted papilloma in the right max-illary sinus. ….. (I) Multilayer centripetal resection with 1-cm margins.”